# High-Rise Residential Outdoor Space Value System: A Case Study of Yangtze River Delta Area

**DOI:** 10.3390/ijerph20043111

**Published:** 2023-02-10

**Authors:** Jing Yang, Yingzhu Liu, Boyi Zhang

**Affiliations:** 1Department of Architecture, School of Architecture, Southeast University, Nanjing 210096, China; 2Department of Architecture, Graduate School of Design, University of Pennsylvania, Philadelphia, PA 19104, USA

**Keywords:** high-rise residential complex, outdoor space, value system

## Abstract

The outbreak of COVID-19 has drawn wider attention from residents with growing demand for outdoor space in residential areas because of restrictions on residents’ mobility, especially in China. However, the high-rise residential complex in China is featured with a high population density along with less outdoor space per household. This means that the current status of outdoor space in residential areas is far from satisfying residents’ growing needs. This is consistent with our preliminary survey that highlights general low satisfaction of residents with outdoor space. According to the hierarchical theory of needs, a literature review, and a questionnaire survey, a framework is proposed in this study to examine the universal value system of high-rise residential outdoor space using the Yangtze River Delta Area as a case study. This framework consists of six dimensions, i.e., space physical comfort (physical environment and space size), space function (functional complexity and scale, age-range, and time-range), space safety (daily, social, and hygiene safety), space diversity (spatial layerings, forms, and scales diversity), accessibility (spatial attraction and concentration and path clarity) and sustainability (cultural, social, ecological, and financial sustainability). Consequently, a questionnaire was designed according to the framework and 251 valid questionnaires were received. Then, structural equation modeling (SEM) was undertaken to examine the impact of each dimension on the value of outdoor space and the framework was optimized into four dimensions, i.e., space physical comfort, space function, space safety, and DAT (space diversity, accessibility, and sustainability). Finally, the mechanism of how outdoor space quality influences the high-rise residential complex is analyzed. These findings provide useful input for the future planning and design of high-rise residential areas.

## 1. Introduction

It is one of the major criteria in urban design to maintain people’s physical and mental health and to stimulate people’s social needs through the nearest public space [1]. As one of the places of daily life, residential outdoor space serves the functions of meeting the needs of residents for rest, entertainment, and neighborhood social interaction. Meanwhile, it plays a crucial role in the physical and mental health of residents. At the same time, housing satisfaction forms an integral part of the overall satisfaction of residents [2]; among all the contributions to housing satisfaction, outdoor space cannot be overlooked.

In China, residential areas are featured with high-rise, high development intensity, high population density, and small residential blocks. In big cities, China’s current residential buildings have plot ratios higher than 2.0 within plot sizes of 5 ha. [3,4]. This has resulted in a high population density along with a relatively greater pressure and challenge of outdoor space in residential areas compared to the residential areas in other countries. Especially, the outbreak of COVID-19 has changed the way of living. Countries, in turn, have introduced policies to prevent the spread of COVID-19. Among them, China has one of the strictest policies that leads to a higher residential dependency on outdoor space. Since the outbreak of COVID-19, China has taken multiple isolation measures to prevent the spread of the virus. This has further increased residents’ attention and dependence on the outdoor space of residential areas. Meanwhile, this isolation has brought the importance of the living environment to residents’ attention. A green and healthy outdoor environment in residential areas can not only meet the fundamental needs of residents’ daily activities but also soothe their bodies and minds and delight their spirits [5]. Meanwhile, it is the primary line of defense for residents to fight the epidemic and ensure safety and health [6]. We undertook a preliminary survey of 240 households in 12 residential areas including Nanjing, Wuxi, and Hangzhou after the pandemic (July 2021–December 2021). It was found that residents’ satisfaction with outdoor space (3.71/5) is lower than indoor space satisfaction (3.92/5). Outdoor space, as a crucial part of residential areas, is currently not satisfactory. In particular, the main dissatisfactions with outdoor space are concentrated in outdoor activity space, green landscape, road design in residential areas, and parking issues (Table 1). In addition, activities were restricted, and residents of all ages had a marked increase in demand for healthy outdoor spaces and activity venues, as well as types of facilities and activity venues for all ages, during the pandemic.

On the other hand, to ensure the quality of outdoor space, relevant policy makers are required to make appropriate planning and design decisions. Therefore, policy makers are critical to ensuring resident satisfaction [7]. However, there are currently no planning-related indicators to control the quality of outdoor space in the Chinese regulations at the national, provincial, and municipal levels. This has seriously affected the effective control of outdoor space quality. Therefore, it is critical to better understand the influencing factors and value system of outdoor space quality in the post-pandemic era to improve the living quality of residents. This ensures the sustainable development of the living environment, and helps developers, planners, and designers for optimal design of outdoor spaces. Furthermore, it provides practical and effective ideas, laying the foundation for the formulation of outdoor space quality-related indicators in the later stage.

## 2. Literature Review

### 2.1. High-Rise Residential Outdoor Space Value Connotation

Value is the benefit relationship in which the object can satisfy the needs of the subject. Meanwhile, it represents a utility relationship between the attributes and functions of the object as well as the needs of the subject. The value of outdoor space in high-rise residential areas is reflected in the fit of spatial attributes and functions with the demands of residents.

According to Maslow’s hierarchy of needs, human needs can be divided into five levels, i.e., physiological needs, safety needs, belonging and love, esteem, and self-actualization [8]. Similarly, the needs of residents in residential areas for outdoor space can be divided into the aforementioned five levels. The outdoor space of the residential area can provide residents with living, rest, recreation, and other living functions, reflecting its material attributes to meet the needs of the residents. The outdoor space of the residential area also guarantees the residents’ safety, provides the residents a sense of belonging, and at the same time ensures the interaction of residents. In addition, the use of the residential outdoor environment along with the residents’ participation in the construction of these spaces not only satisfies the self-realization needs of the residents, but also enables the residential area to develop healthily. In other words, the subject acts on the object, making the object have greater value, from which the value is created by the two.

### 2.2. Residents’ Demands for Outdoor Space

Previous studies have documented residents’ demand for open space, and residents have shown their willingness to pay higher for open spaces [9]. Therefore, the importance of open space to residents is well-recognized in terms of outdoor thermal comfort, spatial function, accessibilities, etc. Accessing open space is crucial for residents across various groups such as elderly, children, people with disabilities, etc. [10]. Green space, as one factor that contributes to open space, is beneficial for human health both physically and mentally [11,12,13,14]. It provides fresh air, is friendly to the human respiratory system, prevents several diseases [15], and provokes interaction between neighbors [12]. Thermally comfortable outdoor environments may facilitate human interaction with the environment, whilst thermally uncomfortable spaces may keep residents indoors and increase energy consumption [16]. Functions of open space including “opportunities for recreation, pleasing views, or absence of negative externalities” [17] provide new possibilities of human activity, although housing has fulfilled a part of residents’ needs [17]. In addition, accessibility ensures the utilization of open space. Open space is especially appreciated during COVID-19 since it is restorative and provides a chance of escaping the pressure of the crisis [18]. Even though the pandemic has almost ended, it has attracted more attention to open space. It is necessary to induce a design philosophy that is capable of being used widely for open space.

### 2.3. The Value Composition of Residential Areas Outdoor Space

In 1961, WHO proposed four living environment concepts: safety, health, convenience, and comfort. Yasushi ASAMI [19] added sustainability indicators based on the WHO living environment concept and carried out evaluations on each value dimension. Du [20] (2002) summarized 12 influencing factors of living environment satisfaction and conducted quantitative analysis. Huang [21] (2006) proposed an evaluation system related to spatial design elements from the perspective of residents–neighborhood communication including paths, nodes, boundaries, etc. Xu [22] (2009) analyzed the ways to enhance the outdoor value from the production cost of residential outdoor space and the composition of customer value including basic functions, landscape, and social and ecological efficiency. Fan [23] (2011) and Dong [24] (2022) found that green spaces in residential areas are useful to release the pressure of residents and also enhance residential mental health by promoting exercise and communication between residents. Sepide [25] (2015) examined the quality influencing factors of residential open space from five perspectives: function, semantics, environment, sociality, and form. Du [26] (2016) carried out a quantitative calculation design of the accessibility and functionality of the activity space. Pavla [27] (2017) established a design system on residential open space evaluation based on sustainable perspectives involving ecology, health and hygiene, sociality, and economics. Shu et al. [28] (2018) explored the influencing factors of residential outdoor space quality from the perspectives of safety, health, comfort, sustainability, and satisfied residents. Eizenberg et al. [29] (2019) proposed outdoor space quality evaluation standards including spatial diversity, accessibility, and greening quality and clarified the calculation methods of each index. In addition, several studies have been conducted on the quality of outdoor activity space from the perspective of space vitality [30,31,32]. These studies explored the value dimensions of outdoor space from a macro perspective, from which the demand for quality of space changes with the development of the times. It provides fresh air friendly to the human respiratory system, prevents several diseases, and promotes the sustainability proposed by WHO. 

In addition, many studies have attempted to analyze a pacific value dimension, such as the material form of space, including greenery form [13,33,34,35,36], housing form [37], special boundary form [38], etc. Some studies present analysis of the public space safety of high-rise residential buildings [39,40]. There are also studies on the connotation and specific composition of space accessibility [41,42,43]. The discussion on spatial sustainability is more extensive, including sociocultural sustainability, ecological sustainability, economic sustainability, etc. [44,45,46,47,48,49]. These studies define each value dimension and its specific composition more accurately, so that the construction of the outdoor space value system of high-rise residential areas is more comprehensive, specific, and precise.

It is difficult to judge the difference in the impact of each value dimension on the quality of outdoor space [33,37,50], and it leads to the difficulty of obtaining the focus of outdoor space design. Most of the quantitative research focuses on a single value dimension, and relevant data was obtained from field observations [21,51,52] or questionnaire surveys [53,54]. Some studies only put forward a theoretical calculation formula without support of empirical data. To address these gaps, this study aims to develop a framework for the value of outdoor space in high-rise residential areas. Similar to Baylan et al. [55], the impact of each value dimension and value item on the quality of outdoor space will be quantified. Consequently, corresponding measures will be proposed to improve the quality of the outdoor space.

## 3. Construction of Outdoor Space Value System in High-Rise Residential Areas

The attributes (i.e., value items) of outdoor space are drawn from the relevant literature on residents’ needs and the value of outdoor space, as well as semi-structured interviews with residents in multiple high-rise residential areas about their satisfaction and needs of outdoor space (Table 1). The selection of these value items is mainly based on the following aspects: Existing research shows its capability of improving the quality of outdoor space in residential areas [19,20,21,22,23,24,25,26,27,28,29,30,31,32,33,34,35,36,37,38,39,40,41,42,43,44,45,46,47,48,49];In line with the universal needs of residents [9,10,11,12,13,14,15,16,17,18];Value items should be common to all types of outdoor space in residential areas (i.e., greenery, activity space, road, overhead layer, pond, and sky court);Necessity to eliminate the influence of the internal design of the space in the later stage, making value system universal;Reflecting the outdoor characteristics of high-rise residential areas;Necessity to ensure the identifiability and operability of the value system following up with the questionnaire.

Finally, six categories, i.e., physical environment performance [19], spatial functionality [22,25], spatial safety [28,39,40], spatial diversity [25,29], spatial accessibility [29,39,40,41], and spatial sustainability [19,27,28,44,45,46,47,48], and 24 sub-items were extracted (Table 2).

## 4. Research Methods

### 4.1. Research Model

The extensive literature review showed that the main methods used in the quantitative research on outdoor space quality in residential areas include regression analysis, linear correlation analysis, analytic hierarchy process, etc. These methods only allow a single measurement index belonging to a single variable and a single calculation path, which is not suitable for research with multiple and complex variables. The complex and diverse dimensions of the value composition of outdoor space belonging to latent variables are difficult to measure directly. However, the value dimension (i.e., each value item under each dimension) can be indirectly measured through observable indicators. Traditional methods cannot deal with these latent variables effectively. By contrast, structural equation modeling (SEM) can deal with both latent and explicit variables. In addition, the structural equation model can select a more suitable model according to the overall fitting degree of different models to the same sample data.

Therefore, the structural equation model is employed in this study (Figure 1) to test the rationality of the composition of the value system through the survey data. At the same time, it obtains the degree of influence of each value dimension and each value item on the quality of outdoor space. The comfort, functionality, safety, spatial organization, accessibility, and sustainability of the physical environment of outdoor spaces in high-rise residential areas are regarded as latent variables that cannot be directly observed, and their constituent factors are taken as explicit variables reflecting these latent variables. The questionnaire items are set accordingly. IBM SPSS Amos24.0 and IBM SPSS Statistics 25 are used for data analysis and model verification. Path analysis is used to test the influence of various latent variables on the quality of outdoor space.

### 4.2. Sample Data Collection

Relevant data on the impact of various value items on the quality of outdoor space were collected in the form of an online questionnaire. Residents living in 15 high-rise residential communities in Nanjing, Suzhou, and Hangzhou were chosen to participate in the survey. QR codes were printed and handed to property management companies of the 15 communities to distribute questionnaires to residents. The contents of the questionnaires include the following two aspects: The basic information of the respondents, including gender, age, occupation, and the scale of the community they live in, etc.;The respondents’ importance evaluation of the 24 value items. On a five-point Likert scale, 1 to 5 represent basically unimportant, somewhat important, moderately important, relatively important, and extremely important.

The survey was conducted from 12 December 2021 to 20 December 2021. A total of 251 valid responses were received, which met the requirements on sample size of statistical analysis (It is generally believed that the ratio of the obvious variables in the SEM analysis to the required sample size is 1:10~1:15. This paper established a questionnaire containing 24 value items, that is, 24 obvious variables, so the estimated sample size is 240~360).

The information of respondents is shown in Table 3. The statistics of the questionnaire results are shown in Table 4.

### 4.3. Sample Data Collection

#### 4.3.1. Exploratory Factor Analysis

Reliability Analysis

To ensure the consistency, judgment, and importance of each value item in all samples, the reliability test is carried out. Cronbach’s α coefficients of physical performance comfort (0.764), functionality (0.767), safety (0.742), spatial diversity (0.811), accessibility (0.771), and sustainability (0.778) are greater than 0.6, indicating that the data have high reliability.

2.Validity Analysis

Content validity and construct validity tests are needed to judge the accuracy and validity of data. Among them, content validity is a test of item logic. The value items set in this study are all based on the relevant literature and transformed into corresponding items. Therefore, the questionnaire can be considered to have content validity.

Construct validity is the dependent variable to test whether the items of the questionnaire can be measured—that is, through the survey data to test whether the composition of each value dimension is accurate. First, KMO and Bartlett sphericity tests were conducted. The KMO value is 0.920, the Bartlett sphericity test chi-square value is 2819.507, and the significance level is 0.000 < 0.001. This indicates that the statistical test of residents’ evaluation of the importance of each value item of outdoor space is significant, and the data are suitable for validity analysis.

Second, exploratory factor analysis (EFA) is carried out to verify whether each value item matches the latent variable represented by the survey data and whether it belongs to the same dimension. Using factor analysis to extract factors, when extracting the six value dimension factors of the null hypothesis, the cumulative variance contribution rate is 64.319%, and the average variance explained in the Peterson research statistics is more than 56.6% [56], indicating that it can better reflect raw data. Using the orthogonal rotation method to extract common factors, the factor composition is slightly different from the assumptions of the original model: the spatial hierarchy, spatial accessibility, and spatial sustainability in the original hypothesis are extracted into a class of factors abbreviated as DAT. Other value items in environmental scale and space physical environment comfort are not classified into one category, indicating that the data of this item are different from the data characteristics of other items in the dimension and cannot represent the physical environment comfort; this value item needs to be removed. The final composition of the revised value factor is shown in Figure 2.

#### 4.3.2. Confirmatory Factor Analysis

Confirmatory factor analysis is undertaken to examine the theoretical relationship between each value dimension and each value item and the quality of outdoor space in high-density residential areas. Correlation analysis was conducted with physical environment comfort, spatial functionality, spatial safety, and DAT (Table 5). Results showed that the correlation between latent variables are higher than 0.6, and *p*-value < 0.001 is significant, indicating that the four latent variables jointly affect one variable—that is, the quality of outdoor space.

A second-order confirmatory factor analysis model was established, and Amos24.0 was used to estimate the parameters of the set model. The calculation results (Table 6) show that all residual estimation coefficients are positive values, and the non-standard path coefficients significant in *p* are all < 0.001, and all standardized factor loadings R are >0.4, which is in line with the traditional standardized estimate cutoff value of 0.4 [57], showing that there are no offending estimates. The calculated composite reliabilities (CR) are all greater than 0.6, and the average variation extraction (AVE) is generally greater than 0.36, which is within the acceptable range and is in line with the standard of Fornell and Larcker (1981) [58], indicating that all aspects of the model have convergent validity.

#### 4.3.3. Model Adjustments

The fitness of the model is tested by seven main references: chi-square value X^2^, degrees of freedom df, chi-square value/degrees of freedom (X^2^/df), Tucker–Lewis index (TLI), comparative fit index (CFI), standardized root mean square residual (SRMR), and root mean square error of approximation (RMSEA). After calculation, among the initial fitting indicators (Table 7), TLI (0.871) and CFI (0.885) were under fitting standard but very close, so it was considered to modify the model. With the help of modification indices offered by IBM SPSS Amos24.0 and through recalculation, all fit indices of the modified model meet the standards (Table 7).

### 4.4. Model Hypothesis Test

After analyzing the structural model, all paths have passed significance tests, and standardized estimates in Table 8, which show the influence degree of each aspect, are all greater than 0.4. In addition, the first-order factors (physical environment comfort, spatial functionality, space safety, and DAT) estimated by the model as the identification of the second-order factors (outdoor space quality) have high standardized estimate (0.808, 0.958, 0.746, and 0.838) in this study. Most of the variance of the first-order factors can be explained by the second-order factors (65.29%, 91.78%, 55.65%, and 70.22%, respectively), which shows that these four factors are a more generalized latent concept signage and, indeed, factors that affect the quality of outdoor spaces.

## 5. Influence Mechanism Analysis of Outdoor Space Quality in High-Rise Residential Areas

The results show spatial accessibility, originally assumed spatial sustainability, and spatial diversity can be explained by the same latent variable, which is defined as DAT in this paper. The value of outdoor space in high-rise residential areas is composed of four categories: physical environment comfort value, spatial functionality value, safety value, and DAT. All four value categories have a positive impact on the overall value of outdoor space.

### 5.1. Result Analysis of Second-Order Model

According to the factor loadings in Table 7, the overall value of space function has the greatest impact on the quality of outdoor space, followed by DAT and physical environment comfort, and the value of spatial safety has a relatively small impact.

The R value of spatial function is 0.958, which indicates that it has a great influence on the quality of outdoor space in high-rise residential areas. The function of space directly affects the use of space, which in turn affects the participation and experience of residents in outdoor space. According to the previous research (Table 1), residents’ dissatisfaction with outdoor space is often related to dissatisfaction with functions. If the outdoor space does not provide the functions required by residents, then the residents are likely to decrease activities in the outdoor space around the residential area. Numbers of residents in high-rise residential areas are higher than others, and relatively few outdoor spaces are provided to them. Therefore, the configuration of space functions must be reasonably considered in the design of outdoor spaces, and the overall quality should be optimized by means of DAT, space physical comfort, and space safety.The R values of DAT and physical environment comfort are both above 0.8, indicating that the impact on outdoor space quality is great, where the impact of DAT is greater than the impact of physical environment comfort. DAT plays an important role in ensuring the uniqueness of residential areas. Due to strict sunshine requirements in high-rise residential areas in the Yangtze River Delta and considering the development of residential areas, the scale of residential areas is miniaturized, and the layout of buildings is limited—the overall outdoor space shape gradually becomes similar. In addition, DAT is closely related to residents’ mental health—through the processing and application of various spatial forms, scales, and regional cultural morphemes, residents’ sense of belonging and identification with the residential area can be enhanced [59], and social cohesion of residential areas can be improved [33]. The physical environment comfort can optimize the space use experience, while a good physical environment can maintain the residents’ continuous demand for outdoor space, thereby maintaining the overall vitality of the residential area. The optimization of DAT not only has a physiological impact on residents, but it also has an important impact on their psychology, while the impact of physical environment comfort on residents is mostly based on physiological aspects. So, the impact of DAT is slightly greater than that of physical environment comfort.The R value of spatial security is slightly lower (R = 0.746). Safety is the most important condition in the living environment [19], and it is a factor that must be considered in the planning and design of outdoor space in high-rise residential areas. However, the impact of safety value on the quality of outdoor space is not as good as the other three observing from the model calculation results. This is somewhat different from general perception. It can be explained from the following aspects: first, safety value is a universal need of people, and its lower threshold as an outdoor space value is already high, so the influence of safety on improving the quality of space is not as obvious as that of other dimensions; second, the influence of safety has a wide range of effects other than its impact on the quality of outdoor space, and therefore, it is not a one-to-one mechanism impact like other values; and third, today’s residential areas are mostly closed so the safety of residential areas can be guaranteed to a certain extent. Therefore, the influence mechanism of safety value on the quality of outdoor space is not as obvious as other value dimensions, which results in the r value being slightly lower.

### 5.2. Result Analysis of First-Order Model

From the perspective of physical environment comfort, the wind environment (Q1_C3) has the greatest impact on the comfort value, and the R value is 0.684, which is the primary consideration for optimizing physical environment comfort of the residential area, followed by the sunshine condition (Q1_C2, R = 0.638), space temperature (Q1_C5, R = 0.605), air quality (Q1_C4, R = 0.528), and acoustic environment (Q1_C1, R = 0.51), which is the least impactful factor. The wind environment in the residential area, which can adjust the temperature of the outdoor space in the residential area and improve the comfort of the human body, is directly related to the human body feeling. It can also partially alleviate the urban heat island effect, accelerate the diffusion of harmful particulate pollutants, etc., and maintain a healthy outdoor environment. The Yangtze River Delta region is cold in winter and hot in summer. In general, the direction of summer wind is different. Managing the layout of buildings in a proper manner cannot only effectively use the summer wind to achieve cooling and ventilation effects, but also resist the winter wind and avoid the generation of “draft wind”, whirlwind, and other adverse wind hazards. This strategy maintains a comfortable space temperature to oppose the defects caused by the disadvantage of sunshine conditions to a certain extent. The optimization of the wind environment in the residential area has multiple impacts, so its impact on physical environment comfort is also the most important. By contrast, the R value of the sound environment value is as small as 0.51, indicating that the impact on physical environment comfort is not significant. Studies have shown that when the ground layer is disturbed by the same noise source, the ground level has more sound-absorbing media (such as trees), the outdoor noise is often lower than the outdoor noise of higher level [60]. The outdoor spaces of residential areas are generally located on the ground floor, and outdoor activities are mainly dynamic activities, so the requirements for the acoustic environment are not so strict.Among the influencing factors of spatial functionality, the R value of functional space scale (Q2_F2) is 0.757, which has the greatest impact on the quality of outdoor space, followed by compound functions (Q2_F1, R = 0.691) and full-aged space functions (Q2_F3, R = 0.647), and the influence of all-weather and full-time (Q2_F4, R = 0.646) has minimum impact. According to previous research, residents’ satisfaction with the number of activity facilities or site size in the residential area is generally low. High-rise residential areas have a large population, resulting in a higher total demand for outdoor space. Outdoor space, especially effective activity space, is relatively limited. It often gives people a feeling of crowding, which makes some residents refuse to exercise in outdoor spaces, resulting in a decrease in overall satisfaction with outdoor spaces. In addition, the scale of the functional space will also affect the compounding degree of the functions in the space. The larger the scale of the functional space, the more functional requirements it can accommodate, and the easier it is to meet the needs of different groups of people. Therefore, priority should be given to ensuring the number of functional spaces in the outdoor space planning of high-rise residential areas, and other aspects should be optimized after satisfying the number, and the later design should be used to meet the needs of all-weather, all-time, and all-aged activities.Among the factors affecting spatial safety, social safety (Q3_S3, R = 0.715) and health safety (Q3_S4, R = 0.715) have the greatest influence on safety value, followed by disaster prevention safety (Q3_S2, R = 0.671), and daily safety (Q3_S1, R = 0.521) has the least influence. Keeping outdoor spaces hygienic can prevent the spread of (micro-) organisms or viruses. It can be assumed that during the pandemic, the health and safety of outdoor spaces in residential areas received attention from residents, especially when outdoor activities are restricted. Outdoor activities in residential areas can keep residents healthy to a certain extent. The R value of daily safety is 0.521, and only 27.4% of the variation of this sign is explained by latent variables, indicating that the linear relationship between the value of spatial safety and the value of daily safety is not as close as that of other value items. The evaluation of the value importance is generally high, indicating that compared to other safety values, the daily safety value is an extremely important and relatively independent value factor. Since a measurement model of structural equation modeling requires at least three observed variables [56], we failed to calculate daily safety as a separate dimension—it should be studied as an independent value dimension in future research. In addition, this study revealed that the importance of the safety value is considered relatively high by residents, but its calculated factor loading is relatively small by comparing the survey data. This does not mean a low importance of the safety value dimension. Rather, its lower limit is relatively high. Therefore, the effect of safety improvement on improving the quality of outdoor space is not as obvious as the improvement of other value dimensions in terms of improving the quality of outdoor space.Among the influencing factors of DAT, the R values of various value items in spatial diversity, spatial attractiveness, and cultural sustainable value all reach above 0.7, indicating that they have a great influence on the quality of outdoor space. In DAT, the value of diversity has the greatest influence. According to the above passages, the outdoor space of high-rise residential areas in the Yangtze River Delta region are often similar, and the space is usually without variations. The design of spatial diversity can well-alleviate this drawback. The R value of outdoor space diversity (Q4_D3) was the largest at 0.749, followed by spatial hierarchy diversity (Q4_D1, R = 0.739), spatial attractiveness (Q5_A1, R = 0.744), spatial cultural sustainability (Q6_T1, R = 0.704), and spatial morphological diversity (Q5_D2, R = 0.707).

(1) The diversity of outdoor spaces can bring the possibility of various activities and bring different vitality to the outdoor space of the residential area. 

(2) The rich spatial layers of the residential area—a transition from public to private—give people a sense of belonging and identity as it is part of the commute route to home.

(3) Spatial attractiveness is one of the driving forces for residents to choose their settlement in the community or to engage in outdoor activities. The more attractive a space is, the more likely residents will choose it, the more active residents will be in that space, and the longer the space will keep its vitality. It is often seen that some venues originally designed as event spaces in the residential area have been gradually abandoned because they have not been popular. Therefore, how to maintain the attractiveness of the space for a long time is also a consideration for the improvement of the quality of outdoor space.

(4) The cultural sustainability of outdoor space and the diversity of spatial forms are important factors to ensure that the residential area has its own characteristics. At present, due to the development of housing industrialization, all residential areas are identical. As one of the important components of the city, housing plays an important role in the expression and inheritance of regional culture. The diversity of spatial forms in residential areas can strengthen the characteristics of spaces and increase the space playfulness, bringing a variety of space experience to the residents.

(5) The R values of clear road streamlines (Q5_A2, R = 0.648), spatial agglomeration (Q5_A3, R = 0.682), social sustainability (Q6_T2, R = 0.682), ecological sustainability (Q6_T3, R = 0.679), and economic sustainability (Q6_T4, R = 0.644) are between 0.6 and 0.7. The impact of the value of outdoor space is slightly smaller, but it should also be considered for improving the quality of outdoor space.

## 6. Conclusions

This study aims to examine the outdoor space of high-density residential areas in the post-COVID-19 context. In essence, high-rise residential areas mean high-density residential areas (i.e., residential areas with high population density) in China. This is a common practice in residential development in China, which presented significant challenges for the quality of outdoor space. 

Through literature review and residents survey, this study systematically examined the quality of outdoor space of high-rise residential areas.

A structural equation model is developed to verify and quantify the value system, and obtain the final outdoor space value system, including four dimensions: physical environment comfort, spatial functionality, spatial safety, and DAT (spatial diversity, accessibility, and sustainability). At the same time, the influence of each value dimension and each value item on the quality of outdoor space is obtained. The results show that the spatial functionality has the greatest impact on quality, followed by DAT and physical environment comfort, and space safety is slightly lower. This study shows the improvement of spatial functionality, DAT, and physical environment comfort has a more significant impact on the improvement of outdoor space quality. The effect result is more direct and obvious. Therefore, the top priority is to improve the outdoor space functions of residential areas, ensuring a reasonable number of functional spaces. Meanwhile, in addition, this study revealed that the importance of the safety value is considered relatively high by residents, but its calculated factor loading is relatively small by comparing the survey data. This does not mean a low importance of the safety value dimension. Rather, its lower limit is relatively high. Therefore, the effect of safety improvement on improving the quality of outdoor space is not as obvious as the improvement of other value dimensions in terms of improving the quality of outdoor space. In addition, attention should be paid to the DAT and physical comfort of outdoor spaces. The emphasis on the diversity of outdoor space is to meet the different needs of residents, especially during COVID-19. For instance, the design of outdoor space should allow for diversity according to a user’s demand, e.g., accommodating needs of both individual and group activities. The optimization of the ventilation and sunlight conditions of the outdoor space is the top priority that needs to be optimized to improve the physical comfort of the space. High-quality outdoor space will likely encourage more physical activities by residents and, hence, improve their health conditions.

Future research opportunity exists to determine the weighting of indicators in the framework developed in this study for the quality of different types of open/outdoor space. More research could be undertaken to evaluate the impact of different factors in the same types of open spaces on space qualities, e.g., the impact on activities by vegetation with the same open space but with different soil layer conditions above parking lots [61]. This will provide useful guidance to developers and designers in the design of the outdoor space of the residential area to more purposefully optimize its quality.

## Figures and Tables

**Figure 1 ijerph-20-03111-f001:**
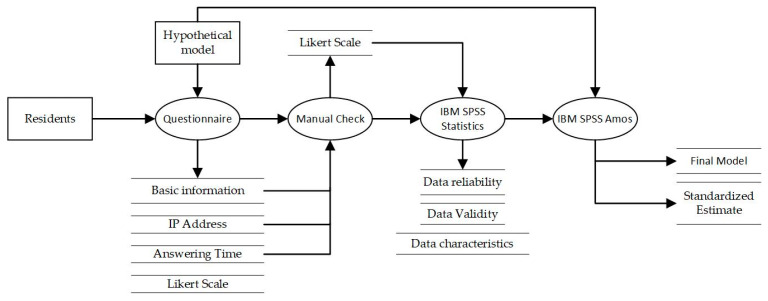
Data flow.

**Figure 2 ijerph-20-03111-f002:**
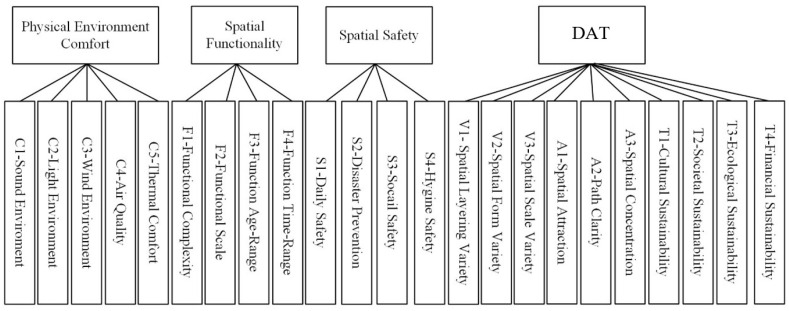
Adjusted value composition of outdoor space.

**Table 1 ijerph-20-03111-t001:** Preliminary Research Summary on Dissatisfactory Outdoor Factors.

Complaint Direction	Complaint Problems	Complaint Times	Total
Outdoor Space Activity Issues	Children’s playground	55	265
	Children’s activity facilities	43	
	Lack of adult resting area in children’s playground	2	
	Indicators being too sharp	2	
	Quality of activity facilities	11	
	Swimming pool being too small and water being too shallow	1	
	Lack of venues for adults and elderly	72	
	Scarcity of venues for adults and elderly	44	
	Distance between venues to residential area	2	
	Scarcity of walking path	2	
	Lack of communication space	3	
	Lack of sunshade and rain shelter	5	
	Lack of resting area	22	
	Venue not concentrated enough	1	
Greenery Issues	Excessive lawn	2	107
	Scarcity of tree canopy	4	
	Lack of flowers	7	
	Scarcity of greenery species	13	
	Lack of activity fields for pets	7	
	Too many mosquitos	32	
	Lack of area signs	1	
	Over simplistic landscape design	26	
	Dim lighting at night	14	
Road and Traffic Issues	Roads and residential buildings are close to each other	2	65
	Inconvenient entrance	9	
	Scarcity of entrance/exit in residential area	10	
	Residents and vehicle entrance/exit not separated	14	
	Excessive ground parking	2	
	Narrow driving path	6	
	Narrow parking space	8	
	Improper parking space setting	1	
	Numbers of charging spaces for non-motor vehicles	6	
	Residents and vehicle path not separated	7	
Others	Community gate impression problem	13	13

**Table 2 ijerph-20-03111-t002:** Value composition of high-rise residential area outdoor space.

Value Dimension	Value Composition	Exposition	Literature Bases
Q1 Physical Environment Comfort	C1-Sound environment	No noise pollution in outdoor spaces	Yasushi (2001) [19]
	C2-Light environment	Main outdoor activity areas have good sunshine conditions without dark and claustrophobic corners	
	C3-Wind environment	Ventilated outdoor space, suitable windspeed, good air flow	
	C4-Air quality	Fresh air in outdoor space without hazardous substances and odors	
	C5-Thermal Comfort	Comfortable temperature	
	C6-Outdoor space size	Appropriate outdoor space size, not too crowded nor too empty	
Q2 Spatial Functionality	F1-Functional complexity	Space has multiple functions instead of one single function	Xu (2009) [22], Saiedlue et al. (2015) [25]
	F2-Functional scale	Number of functional spaces and size of each space meet residential needs	
	F3-Function age-range	Spatial facilities meet the needs of people of all ages	
	F4-Function time-range	Outdoor space satisfies the needs of residents in various climatic conditions and time periods	
Q3 Spatial Safety	S1-Daily safety	Outdoor space ensures the safety of residents’ daily activities, no hidden dangers such as falling objects, collapse, and traffic safety.	Yasushi (2001) [19], Shu et al. (2018) [28], Chen & Wu (2010) [39], Frances et al. (1998) [40]
	S2-Disaster prevention	Outdoor space has certain disaster prevention functions, such as: resisting and alleviating natural disasters, preventing the spread of fire, and preventing the spread of disease, etc.	
	S3-Social safety	Outdoor space ensures social safety, without hidden danger of fighting, theft, abduction, and other illegal and criminal acts	
	S4-Hygine safety	Outdoor space is hygienic, without infectious diseases mediated by microorganisms	
Q4 Spatial Diversity	D1-Spatial layering diversity	Outdoor space has rich spatial transitions, ensures residents’ sense of belonging and identity	Saiedlue et al. (2015) [25], Efrat et al. (2019) [29]
	D2-Spatial form diversity	Various forms of space (enclosed space, linear space, open space, semi-enclosed space) without stereotypes	
	D3- Outdoor space, diversity	Open space with various scales with places suitable for both group and individual activities	
Q5 Spatial Accessibility	A1-Spatial Attraction	Outdoor spaces are visible and attractive to residents	Efrat et al. (2019) [29], Chen et al. (2007) [41], Li & Lu (2005) [42], Peng (2017) [43]
	A2-Path clarity	Streamline organization of the roads is clear and concise, the destination can be easily reached	
	A3-Spatial concentration	Layout of each functional space is relatively concentrated and convenient to enter	
Q6 Spatial Sustainability	T1-Cultural sustainability	Outdoor space reflects regional cultural characteristics	Yasushi (2001) [19], Pavla & Maxmilian (2017) [27], Shu et al. (2018) [28] Zhang et al. (2019) [44], Shen (2019) [45], Efrat & Yosef (2017) [46], Li & Wu (2005) [47], Ebbesen et al. (1976) [48]
	T2-Societal sustainability	Community atmosphere is harmonious and promote a stable development of the neighborhood relationship	
	T3-Ecological sustainability	Outdoor space is ecologically balanced, with healthy environment, diverse, and ecosystem, and rationally used resources	
	T4-Financial sustainability	The resources in the outdoor space are planned reasonably in the early stage and used in the later stage without any waste	

**Table 3 ijerph-20-03111-t003:** Basic information of respondents.

Basic Information		Counts	Proportion
Gender	Male	124	49.4%
	Female	127	50.6%
Age	≤17	3	1.2%
	18–25	29	11.6%
	26–45	72	28.7%
	46–59	140	55.8%
	≥60	7	2.8%
Occupation	Government department	27	10.8%
	State-owned enterprise	90	35.9%
	Private enterprise	64	25.5%
	Social organization	5	2.0%
	Self-employment	15	6.0%
	Others	50	19.9%
Heights of the buildings in the community residents live in *	≤11 storeys	141	56.2%
	12–18 storeys	41	16.3%
	≥19 storeys	39	15.5%
	Hybrid (≤11 storeys & ≥19 storeys)	30	12.0%

* The classification of building floors is generally applied in Chinese real estate industry.

**Table 4 ijerph-20-03111-t004:** Statistics of questionnaire results.

	Number of Valid Cases	Minimum	Maximum	Average	Standard Error of Mean	Variance	Skewness	Standard Error of Skewness	Kurtosis	Standard Error of Kurtosis
Q1_C1	254	1	5	4.25	0.056	0.789	−1.378	0.154	1.902	0.306
Q1_C2	254	1	5	4.10	0.053	0.706	−1.087	0.154	1.634	0.306
Q1_C3	254	1	5	3.96	0.051	0.659	−0.477	0.154	0.010	0.306
Q1_C4	254	1	5	4.57	0.046	0.526	−2.052	0.154	4.794	0.306
Q1_C5	254	1	5	3.71	0.058	0.841	−0.699	0.154	0.644	0.306
Q1_C6	254	1	5	3.79	0.060	0.893	−0.750	0.154	0.468	0.306
Q2_F1	254	1	5	3.95	0.057	0.825	−1.091	0.154	1.473	0.306
Q2_F2	254	1	5	3.88	0.051	0.658	−0.322	0.154	0.047	0.306
Q2_F3	254	1	5	3.92	0.060	0.909	−0.837	0.154	0.610	0.306
Q2_F4	254	1	5	3.46	0.077	1.473	−0.490	0.154	−0.642	0.306
Q3_S1	254	1	5	4.75	0.037	0.335	−3.036	0.154	11.941	0.306
Q3_S2	254	1	5	4.25	0.057	0.829	−1.256	0.154	1.304	0.306
Q3_S3	254	1	5	4.25	0.058	0.831	−1.198	0.154	1.092	0.306
Q3_S4	254	1	5	4.28	0.052	0.668	−1.229	0.154	1.756	0.306
Q4_D1	254	1	5	3.76	0.061	0.929	−0.793	0.154	0.577	0.306
Q4_D2	254	1	5	3.50	0.064	1.035	−0.431	0.154	−0.096	0.306
Q4_D3	254	1	5	3.61	0.064	1.040	−0.563	0.154	−0.093	0.306
Q5_A1	254	1	5	3.74	0.062	0.953	−0.840	0.154	0.618	0.306
Q5_A2	254	1	5	3.87	0.058	0.856	−0.661	0.154	0.248	0.306
Q5_A3	254	1	5	3.53	0.064	1.042	−0.308	0.154	−0.259	0.306
Q6_T1	254	1	5	3.34	0.069	1.201	−0.391	0.154	−0.263	0.306
Q6_T2	254	1	5	3.57	0.059	0.870	−0.592	0.154	0.344	0.306
Q6_T3	254	1	5	3.85	0.056	0.777	−0.688	0.154	0.427	0.306
Q6_T4	254	1	5	3.92	0.060	0.898	−0.693	0.154	0.029	0.306

**Table 5 ijerph-20-03111-t005:** Correlation Analysis.

			Correlations	Estimate	Standardized Estimate	C.R. (t-Value)	*p*
Physical Environment Comfort	↔	Spatial Functionality	0.731	0.254	0.044	5.741	***
Physical Environment Comfort	↔	Spatial Safety	0.737	0.199	0.035	5.744	***
Physical Environment Comfort	↔	DAT	0.588	0.154	0.029	5.26	***
Spatial Functionality	↔	Spatial Safety	0.672	0.326	0.054	6.016	***
Spatial Functionality	↔	DAT	0.833	0.391	0.061	6.361	***
Spatial Safety	↔	DAT	0.6	0.219	0.039	5.65	***

↔ means correlation relationship. *** means *p*-value < 0.001, calculation has passed significance test.

**Table 6 ijerph-20-03111-t006:** Measurement model validity analysis.

Latent Variable	Observed Variable	Standardized Estimate (R)	*p*	C.R.	AVE
Outdoor Space Quality	Physical Environment Comfort	0.804		0.905	0.706
	Spatial Functionality	0.962	***		
	Spatial Safety	0.742	***		
	DAT	0.837	***		
Physical Environment Comfort	Q1_C2	0.638		0.732	0.356
	Q1_C3	0.689	***		
	Q1_C4	0.521	***		
	Q1_C1	0.508	***		
	Q1_C5	0.607	***		
Spatial Functionality	Q2_F4	0.652		0.78	0.472
	Q2_F3	0.638	***		
	Q2_F2	0.754	***		
	Q2_F1	0.697	***		
Spatial Safety	Q3_S4	0.72		0.748	0.43
	Q3_S2	0.654	***		
	Q3_S3	0.71	***		
	Q3_S1	0.519	***		
DAT	Q6_T4	0.611		0.907	0.495
	Q6_T3	0.662	***		
	Q6_T2	0.698	***		
	Q6_T1	0.709	***		
	Q5_A3	0.701	***		
	Q5_A2	0.668	***		
	Q5_A1	0.737	***		
	Q4_D3	0.74	***		
	Q4_D2	0.731	***		
	Q4_D1	0.765	***		

*** means *p*-value < 0.001, calculation has passed significance test.

**Table 7 ijerph-20-03111-t007:** Model Fit Indices.

Model Fit Indices	X^2^	df	X^2^/df	TLI	CFI	SRMR	RMSEA (LO,HI)
Fitting Standard	As small as possible	/	<3	>0.9	>0.9	<0.08	<0.08
Original Model Fitting	512.736	226	2.269	0.871	0.885	0.053	0.071 (0.063,0.079)
Modified Model Fitting	378.149	218	1.735	0.926	0.936	0.048	0.054 (0.045,0.063)

**Table 8 ijerph-20-03111-t008:** Standardized Regression Weights.

Second-Order Model				
First-Order Latent Variable		Second-Order Latent Variable	Standardized Estimate (R)	*p*
Physical Environment Comfort	←	Outdoor Space Quality	0.808	
Spatial Functionality	←	Outdoor Space Quality	0.958	***
Spatial Safety	←	Outdoor Space Quality	0.746	***
DAT	←	Outdoor Space Quality	0.838	***
**First-Order Model**				
**Observed Variable**		**First-Order Latent Variable**	**Standardized Estimate (R)**	** *p* **
Q1_C3	←	Physical Environment Comfort	0.684	***
Q1_C2	←	Physical Environment Comfort	0.638	
Q1_C4	←	Physical Environment Comfort	0.528	***
Q1_C1	←	Physical Environment Comfort	0.51	***
Q1_C5	←	Physical Environment Comfort	0.605	***
Q2_F4	←	Spatial Functionality	0.646	
Q2_F3	←	Spatial Functionality	0.647	***
Q2_F2	←	Spatial Functionality	0.757	***
Q2_F1	←	Spatial Functionality	0.691	***
Q3_S4	←	Spatial Safety	0.715	
Q3_S2	←	Spatial Safety	0.671	***
Q3_S3	←	Spatial Safety	0.715	***
Q3_S1	←	Spatial Safety	0.521	***
Q6_T4	←	DAT	0.644	
Q6_T3	←	DAT	0.679	***
Q6_T2	←	DAT	0.682	***
Q6_T1	←	DAT	0.704	***
Q5_A3	←	DAT	0.682	***
Q5_A2	←	DAT	0.648	***
Q5_A1	←	DAT	0.744	***
Q4_D3	←	DAT	0.749	***
Q4_D2	←	DAT	0.707	***
Q4_D1	←	DAT	0.739	***

← means regression path. *** means *p*-value < 0.001, calculation has passed significance test.

## Data Availability

The date that supports the finding of the study are available on request from the corresponding author [Y.L.], upon reasonable request.

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
