# Peer review of "High-Rise Residential Outdoor Space Value System: A Case Study of Yangtze River Delta Area"

_ijerph, 2023, doi:10.3390/ijerph20043111_

Round 1

Reviewer 1 Report (Previous Reviewer 1)

It seems that the Authors still have not understood the concepts and categories cited in particular in the articles Saiedlue et al. (2015) and Efrat et al. (2017). The category of aesthetic quality includes, among others, the subcategory of diversity (and not vice versa). The aesthetic values also include the readability of the system and its attractiveness. Accessibility is a type of functionality. (cf. Saiedlue et al. (2015) Fig. 6 – semantic and form factor)

This part of the study needs to be corrected.

Author Response

Dear Reviewer, 

Thank you for your helpful comments. We've modified our manuscript accordingly. Below is the response:

Comment: It seems that the Authors still have not understood the concepts and categories cited in particular in the articles Saiedlue et al. (2015) and Efrat et al. (2017). The category of aesthetic quality includes, among others, the subcategory of diversity (and not vice versa). The aesthetic values also include the readability of the system and its attractiveness. Accessibility is a type of functionality. (cf. Saiedlue et al. (2015) Fig. 6 – semantic and form factor) This part of the study needs to be corrected

Response: Thank you for pointing this out. These papers have been carefully reviewed. Consequently, category of “spatial aesthetic diversity” is corrected to “spatial form diversity”. The corresponding explanation in the table has been elaborated more based on the change.

Reviewer 2 Report (Previous Reviewer 2)

The manuscript has been significantly improved 

Author Response

Dear reviewer:

Thank you for your encouragement.

Reviewer 3 Report (New Reviewer)

The article titled "High-rise Residential Outdoor Space Value System: A Case Study of Yangtze River Delta Area" is a meticulously prepared article on a very necessary subject. It can be said to be an original choice in terms of both the chosen subject and the field of application. However, I will make small suggestions to improve the quality of the article.

Author Response

Dear Reviewer.

Thank you for your helpful comments. We've modified our manuscript accordingly. Below are our responses:

  1. Comment: The abstract should actually reflect the entire article. In this respect, the summary should include how many people were questionnaire surveyed.

Response: Thanks for your suggestion. Relevant information has been added in the abstract.

  1. Comment: In addition, in a very short way, the factors that are important from the four dimensions: space physical comfort, space function, space safety and DAT (space diversity, accessibility, sustainability) should be included.

Response: Thanks for pointing out. Relevant factors have been included in the abstract.

  1. Comment: However, in the section of: 4. Research Method, 4.2. Sample Data Collection: It would be more meaningful to write clearly in which region people participated in the questionnaire questionnaire survey conducted. For example, were people questionnaire surveyed randomly living in any high-rise residential complex or people living in settlements in the Yangtze River Delta Area?

Response: Thanks for the suggestion. 15 high-intensity residential communities in Nanjing, Suzhou, Hangzhou. were chosen to participate the survey. QR codes were printed and handed to property management companies of the 15 communities to distribute the questionnaire to residents. These details are added to the paper.

  1. Comment: In addition, the article titled "SENSE OF PLACE AND SATISFACTION WITH LANDSCAPING IN POSTEARTHQUAKE HOUSING AREAS: THE CASE OF EDREMIT TOKI-VAN (TURKEY)" can also be used in the literature. For example, a comparison can be made about the different approach of outdoor satisfaction measurement in this article (Same method is used in this article).

Response: Thanks for the suggestion. We have carefully compared these two articles, and the recommended article has been cited.

This manuscript is a resubmission of an earlier submission. The following is a list of the peer review reports and author responses from that submission.

Round 1

Author Response

Dear reviewer, 

We thank you for your invaluable review for our manuscript. Please see the attachment for our responses.

Thank you

Reviewer 2 Report

Authors have prepared very interesting manuscript. It points to importance of outdoor places especially during the covid crises which some countries experienced harder than others. So, China is such exmple.

Authors gave very detailed insight into literature that deals with perception of outdoor places. Methods are clear, and results are expected. I could suggest that authors add possibilities for implementation od the results in the planning system, with emphasize on international context of the results.

Author Response

Dear reviewer, 

We thank you for your invaluable review for our manuscript. The conclusion has been modified according to your suggestion.

Thank you

Reviewer 3 Report

Dear Authors,

The study is interesting but the way it is currently written doesn’t allow for replicability. This needs to be corrected.

For instance, much more details on the data collection would need to be provided: how was the questionnaire distributed ('online' is superficial), basic demographics data, ethics review etc.

Moreover, it is difficult to review the modelling work results without being able to access the data.

The data flow is not described in sufficient detail. It would be good to adhere to a protocol for reporting structural equation models.

Terms ‘spatial functionality’ and ‘the functional value of space’ cannot be used interchangeably, please rephrase carefully.

Including the size of the plot and building density of the area would make more sense than focusing on the height only. Please consider additional variables, otherwise you are not telling the full story.

How is a high-rise area defined? Is it about the number of storeys or does your selection criteria take into the account building footprint ratios and building plot ratios?

The use of the land below the public space (such as parking) is another major factor influencing the quality of the vegetation. Does your framework account for such factors? If not, please discuss.

Line 442 - please check date or data?

Lines 99-100 Mentions of the 'post covid era' are irrelevant for this study, please remove. The values this study is focusing on were equally important before the COVID outbreak.

Lines 456 – 457 – ‘"quantity" of outdoor space quality – please rephrase or consider replacing with ‘index of outdoor space quality’ or ‘measure of outdoor space quality’

Abstract should mention structural equation modelling as that is the main part of this work.

Conclusion mainly talks about safety and reads more like a discussion

There are many errors in the list of references. 

Author Response

(The authors gave the same response as above.)

Round 2

Reviewer 1 Report

Review of article improvement

The Authors have improved the text and clarified most of the doubts.

Comments still concern the addition of the introductory part. Characteristics based on specific indicators were expected. The statements made (vv. 34-43) are awkward and contribute little real information.

The reviewer maintains, despite the explanations received, the claim that the wording of the questionnaire is to some extent biased and affects the nature of the results obtained. The Authors' answers still have not dispelled these doubts. Therefore, I propose that the text should clearly indicate the literature bases for categories formulated in this way. The bases quoted above formulate them slightly differently.

Reviewer 3 Report

Dear Authors,

Thank you for improving the text following my advice. However, I believe the replicability of the study should still be improved. It would be most useful to make the dataset available or to provide clearer information about it.

Line 16 – it would be important to specify whose value system would this be – property agents, residents, architects, planners, universal? Moreover, this is related to my second concern – the 24 concepts used in the survey are sometimes very close to architectural jargon and it is not clear how you made sure that the participants have all understood those questions in an unified way? This is potentially a major issue. The data checks you have included do not prove that the participants share the understanding of those concepts with the authors.

Thank you for providing more data about the survey but this is still not sufficient. International readers would need more information about the platform used, more info would be needed on how were the QR codes distributed, how were the participants giving their consent, were they reimbursed, etc. It would be useful to have the questionnaire added as an appendix, in the original language and the English translation. Moreover, you have not replied to my query if the research design was reviewed by an ethics committee, and this is serious concern given that your study was dealing with human subjects.

Further, I didn’t find Figure 1 to be contributing to clarity as is not describing the data flow but rather your methodological framework, which was already clear. A detailed description of data flow is crucial to understand your rationale behind the particular analytical strategy.

Thank you for the explanations given about the density and the thickness of the layer of soil above the parking facilities. I believe this should be added to the discussion about the limitations of the study, with suitable references.

Line 210 – relevant literature is mentioned but not cited, this is a serious issue. There are many vague points in this subsection, all questioning the reliability of your initial study design. For instance – line 217 – how was this checked and confirmed? Line 218 – vague category. Lines 224 – 226 – how was this extraction performed?

Line 255 – does Amos24.0 refer to IBM SPSS Amos? Please cite both software packages correctly. I suppose those would be IBM SPSS Amos and IBM SPSS Statistics.

Line 260 – still date instead of data, please do check this more thoroughly.

Line 264 – perhaps high density instead of intensity? Further, the area of your study covers around 100 000 square kilometres, and a population of more than 100 million residents. How did you reach the 250 participants while making sure they live in a type of an area that is relevant for your study in terms of urban mophology? Does the ‘scale of the community they live in’ provide a sensible proxy for that in some way?

Line 271 – what qualifies as a valid questionnaire?

Table 3 – it is great to read about the number of storeys. It is unclear what a hybrid would mean in this context though. Please explain more clearly.

Lines 288 – 293 – it would be useful to report the coefficients more clearly so to mention the number of each subscales (physical performance, comfort, etc.), where the coefficient could be in the parentheses.

Table 7 – I am not sure how to read this table. Perhaps a line could be added in 4.4. briefly explaining the meaning of the way the standardised regression weights are reported? Does this mean that both Q1-6 levels and the 24 sublevels were somehow assessed independently so a weighting at the Q level is higher than at a sublevel?

Line 372 – I don’t understand how this can be concluded from Table 6. Please revise.

Line 375 – Does this refer to the spatial functionality in the Table 7? I understand that data for this was obtained from Q2, F1-F4, but looking at the Table 2, it is unclear how were all the components measured using an online survey. For instance, it is not clear if an average participant could reply in a meaningful way to a question such as ‘how important it is for you that spatial infrastructures meet the needs of people of all ages?’ One of the issues could be that an average participate doesn’t understand the meaning of the term ‘spatial infrastructure’.

Line 394 – rephrase ‘unindectical’

Line 412 – perhaps ‘treshold’ instead of ‘limit’? Otherwise, this is a very clear section. Indeed, this prompts for replication of the study in countries where outdoor safety is managed in a different way.

5.2. – are all of those based on the Table 7? Please provide more clarity throughout the text when referring to specific numerical values.

Line 435 – perhaps ‘eddy current’ could be rephrased, as it might not be the most straightforward term when describing the wind-related phenomena.

Line 514 – how were the spatial scales measured? The questionnaire seems to be talking about over crowdedness of outdoor space. Spatial scale reads as a wrong term here. Moreover, a number of people per square meter in a park could have a vastly different outcome in terms of crowdedness if you would be assessing a park, a square, or a street. Please rethink the terms you are using. What were the original terms used in the questionnaire? This conclusion is very vague, indeed.

Lines 516- 519 – this reads vague and is not clear how it is supported by the results.

Line 523 – this doesn’t seem to be supported by the results, how is the lifespan of a space measured within this study?

Lines 553 – 555 – it would be expected that this ranking works differently after certain thresholds are met. The results should be interpreted having in mind that not all of those measures can be measured equally, some would be essential and some less so.

Line 558 – what previous period does this refer to? Please add a reference if appropriate.

Line 562 – the effect of demographics or membership to a group has n

ot been tested so this statement reads too general and obsolete.

Line 597 – this is a clear COVID – related conclusion, which is great but it is not clear if the results truly support this as demographics were no included in the model. Only 2 F3 was used as a proxy but this should have been confirmed by looking into the demographics data. Moreover, if you want to push for the COVID-related aspects of this study, the key would be to describe the COVID-related context surrounding your data collection campaign. For instance, it should be clear what have your participants experience before and during the data collection in terms of COVID-related policies and behavioural patterns.